# In-Context Learning Under Regime Change

## Abstract

Non-stationary sequences arise naturally in forecasting and decision-making. The data-generating process shifts at unknown times, and models must detect the change, discard or downweight obsolete evidence, and adapt to new dynamics on the fly. Transformer-based foundation models increasingly rely on in-context learning for time series forecasting, tabular prediction, and continuous control. As these models are deployed in non-stationary environments, understanding their ability to detect and adapt to regime shifts is important. We formalize this as an in-context change-point detection problem and formally establish the existence of transformer models that solve this problem. Our construction demonstrates that model complexity, in layers and parameters, depends on the level of information available about the change-point location, from no knowledge to knowing exact timing. We validate our results with experiments on synthetic linear regression, where trained transformers match the performance of optimal baselines across information levels. We also show that encoding and incorporating changepoint knowledge indeed improves the real-world performance of a pretrained foundation models on infectious disease forecasting and on financial volatility forecasting around Federal Open Market Committee (FOMC) announcements without retraining, demonstrating practical applicability to real-world regime changes.

## 1. Introduction

Non-stationary environments are the norm in real-world sequential estimation. In control systems, component failures or environmental changes induce abrupt shifts in dynamics (1; 2). In the Markov jump linear system (MJLS) framework (3), such shifts are modeled as transitions of a discrete mode variable, with estimation quality depending on knowledge of the switching signal. Similar phenomena arise in financial markets, where earnings reports and policy announcements shift volatility regimes (4; 5), and in epidemiology, where interventions alter transmission dynamics mid-outbreak (6). A common challenge is change-point detection and adapta-

tion: the data-generating process shifts at an unknown time, requiring estimators to detect the change, discard obsolete evidence, and adapt online.

Classical approaches to change-point detection are well studied. CUSUM procedures (7; 8) provide sequential tests with optimality guarantees, while Bayesian online change-point detection (9) maintains a posterior over run lengths for principled adaptation. However, these methods typically require explicit model assumptions and are designed for a single detection task.

In parallel, machine learning has shown that transformers can solve estimation problems in-context, adapting from prompt examples without parameter updates (10; 11; 12). Theoretical work shows that transformer forward passes can implement gradient descent, least-squares regression, and Bayesian model averaging (13; 14; 15; 16; 17). These ideas underpin foundation models for time series (18; 19), tabular prediction (20), and other sequential tasks where non-stationarity is common. Yet most in-context learning theory assumes stationary prompts from a single task; the closest related work (21) considers heterogeneous prompts but not temporal regime shifts.

This assumption is increasingly misaligned with practice. As foundation models are deployed in non-stationary environments, understanding their ability to detect and adapt to regime shifts becomes critical—not only from a machine learning perspective, but also from the classical viewpoint of the required information structure.

**Contributions.** We formalize in-context change-point detection as a framework for studying transformer adaptation to non-stationary sequences. Our contributions are: **(i) Problem formulation.** We introduce a family of piecewise-linear in-context learning problems with an unknown change point and consider a hierarchy of information levels—from no knowledge to exact timing—showing how positional encoding can convey this side information. **(ii) Constructive theory.** We provide explicit transformer constructions that solve the in-context change-point detection and adaptation problem, and show that the required model complexity (depth and width) depends on the available side information, establishing a capability–complexity tradeoff. **(iii) Synthetic validation.** We train GPT-2–style transformers on piecewise-linear regression tasks with stochastic change points, and show that they match theoretically optimal base-

lines, demonstrating implicit in-context change-point detection and adaptation. **(iv) Real-world validation.** We show that our positional encoding methods improve a pretrained time series foundation model on infectious disease forecasting and financial forecasting around Federal Open Market Committee (FOMC) announcements, without retraining, demonstrating practical gains for deployed models.

## 2. Transformers Can Implement Bayesian Change-Point Adaptation

We show that a causal transformer can approximate the Bayesian model-averaged (BMA) predictor for the piecewise-linear change-point problem defined in (1), and characterize how model complexity depends on available side information.

### 2.1. Problem Setup

We observe $N$ input-label pairs $\{(x_i, y_i)\}_{i=1}^N$ generated by two linear regression tasks separated by change point $n_1^*$:

$$y_i = \begin{cases} \langle w_1, x_i \rangle + \varepsilon_i, & i \leq n_1^*, \\ \langle w_2, x_i \rangle + \varepsilon_i, & i > n_1^*. \end{cases} \quad (1)$$

We assume bounded inputs: $\|x_i\| \leq B_x$ and $|y_i| \leq B_y$ for known constants. The noise is Gaussian, $\varepsilon_i \sim \mathcal{N}(0, \sigma^2)$, and we place a Gaussian prior $w_1, w_2 \sim \mathcal{N}(0, \lambda^{-1} I_d)$ on the regression weights. The change point satisfies $n_1^* \in \mathcal{K}_t \subseteq \{1, \ldots, t-1\}$, where $\mathcal{K}_t$ is the set of admissible change-point locations at time $t$, with $|\mathcal{K}_t| \leq R$. We assign a uniform prior $\pi_k = 1/|\mathcal{K}_t|$ over candidates.

### 2.2. Main Result

BMA is the optimal prediction strategy under squared error loss when the true model lies within the hypothesis class (22; 23). Recent work has shown that transformers implicitly implement BMA for stationary in-context learning (16); our construction extends this to the non-stationary setting with an unknown change point. The BMA predictor maintains a posterior over change-point hypotheses and combines their predictions:

$$\hat{y}_t^{\text{BMA}} = \sum_{k \in \mathcal{K}_t} \alpha_k(t) \, m_k(t), \quad (2)$$

where $m_k(t) = x_t^\top \mu_k$ is the posterior predictive mean under hypothesis $k$ and $\alpha_k(t) \propto \pi_k \, p(Z_{<t} \mid k)$ is the posterior change-point weight. Under the Gaussian prior and noise model, both quantities depend on the sufficient statistics $A_k(t) = \sum_{i=k+1}^{t-1} x_i x_i^\top$, $b_k(t) = \sum_{i=k+1}^{t-1} x_i y_i$, and $c_k(t) = \sum_{i=k+1}^{t-1} y_i^2$ for the post-change segment under hypothesis $k$.

Our construction (in appendix) shows that a causal transformer can compute these quantities and approximate (2).

**Theorem 1** (Approximation of Change-Point BMA). *Consider the piecewise-linear model* (1) *with bounded inputs and the Gaussian prior and noise model specified above. For every $\epsilon > 0$, there exists a causal transformer $T_\epsilon$ such that*

$$\left| T_\epsilon(Z_{<t}, x_t) - \hat{y}_t^{\text{BMA}} \right| \leq \epsilon \quad (3)$$

*for all $t = 1, \ldots, N$ and all input sequences satisfying the boundedness assumptions. The number of layers is constant for fixed $\epsilon$, and the number of attention heads and hidden dimension scale with $|\mathcal{K}_t|$ and $d$.*

The proof is given in the appendix.

## 3. Synthetic Experiments

We evaluate our theoretical framework using piecewise-linear regression. A sequence is generated by one linear process up to an unknown change point and by a different linear process afterward. The transformer must detect the transition from context and quickly adapt its predictions to the new regime. We train GPT-2 style causal transformers under stochastic change-point locations and compare them to ideal baselines matched to the information available to the model. Our goal is not only to test whether transformers benefit from additional change-point information, but also whether they approach the best possible in-context strategy under each information regime.

**Encoding Change-Point Information** We use positional encoding (PE) to provide side information about the change point. This lets us vary the amount of information available to the model without changing the architecture or training objective. We consider three PE schemes—no PE, sinusoidal PE, and linear PE—under four information levels: **(i) no information:** the model receives no explicit signal about the change point. **(ii) Support known:** the model is told only the interval in which the change point lies. **(iii) Known in advance:** the exact change point is encoded before the transition occurs. **(iv) Known afterward:** the model is informed only after the transition has occurred. This hierarchy allows us to study how side information changes the difficulty of in-context adaptation.

**Setup** We consider a piecewise-linear regression task. Each prompt consists of $N = 30$ data-label pairs generated by one linear predictor before the change point and a different linear predictor afterward. During training, the change point is drawn uniformly from $\{10, \ldots, 20\}$. At test time, we fix the change point to $n_1 = 12$ so that performance cannot be explained by a simple bias toward the midpoint of the training distribution.

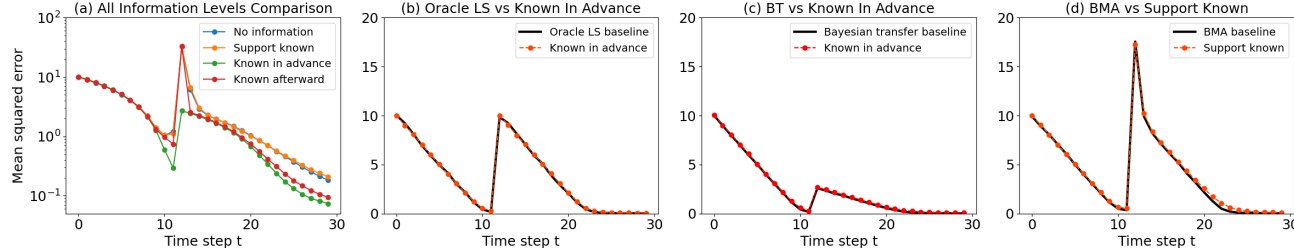

*Figure 1.* Piecewise-linear regression results. Mean squared error averaged over 5,000 test trajectories as a function of time step $t$, with the change point at $n_1^* = 12$ (vertical dashed line). **(a)** Comparison of all four information levels on a log scale. Before the change point, the known-in-advance variant achieves the lowest error by isolating pre-change data. All variants spike at the transition; the no-information and support-known variants recover more slowly than the informed variants. **(b)** The known-in-advance transformer closely matches the oracle least-squares baseline, which knows the true change point and fits only on data from the active regime. **(c)** Under a transfer task where $w_2 = -w_1 + \varepsilon$, the known-in-advance transformer matches a transfer ridge regression baseline that leverages pre-change data through the prior, demonstrating that the model learns to exploit the relationship between regimes rather than simply discarding old data. **(d)** The support-known transformer closely tracks the Bayesian model averaging baseline, confirming that transformers with partial change-point information learn to perform approximate posterior averaging over candidate change-point locations in-context.

**Baselines** We compare transformer performance to three ideal baselines. **(i) Oracle ridge regression:** for settings where the model knows the change point, we compare against an oracle that knows the regime boundaries and performs ridge regression using only samples from the active regime. **(ii) Transfer ridge regression:** for the correlated-task setting where $w_2 = -w_1 + \varepsilon\eta$ with $\eta \sim \mathcal{N}(0, I)$, simply discarding pre-change data is suboptimal because the task 1 samples carry information about $w_2$. This baseline first estimates $w_1$ via ridge regression on the pre-change data, obtaining the posterior $w_1 \mid X_1, Y_1 \sim \mathcal{N}(\hat{w}_1, \Sigma_1)$. Propagating through the known relationship gives an informative prior $w_2 \sim \mathcal{N}(-\hat{w}_1, \Sigma_1 + \varepsilon^2 I)$ on the post-change weights, which is then updated via ridge regression as task 2 samples arrive. **(iii) Bayesian model averaging (BMA):** for settings in which the model is not given the change point, we compare against a Bayesian baseline that maintains a posterior over all candidate change-point locations and combines the corresponding ridge regression predictors according to their posterior weights. These baselines represent the ideal prediction strategies with and without change-point information, respectively.

**Results** Figure 1 shows the per-step MSE for the piecewise-linear regression task, averaged over 5,000 test trajectories with the change point fixed at $n_1^* = 12$. Panel (a) compares all four information levels on a log scale. Before the change point, the known-in-advance variant achieves the lowest error, since the other variants must hedge against a possible regime shift, sacrificing pre-change accuracy for robustness. In contrast, knowing the transition occurs at $t = 12$ allows the model to fully commit to the pre-change task. At the transition, all variants exhibit error spikes of different magnitudes. The known-in-advance variant quickly returns to baseline error: it avoids stale dynamics but must initially predict from an uninformed prior. The other three

variants, still relying on pre-change dynamics, incur much larger errors—nearly an order of magnitude higher. After one step, the known-afterward variant receives the change-point signal and immediately matches the known-in-advance performance. The no-information and support-known variants, lacking exact timing, require several post-change observations to detect the shift and downweight outdated data. Their similar recovery reflects that the training distribution already reveals the change-point support $10, \ldots, 20$, making them effectively equivalent.

Panels (b)–(d) compare transformer variants to their corresponding ideal baselines. The known-in-advance transformer closely matches the oracle least-squares baseline (panel b), which uses only data from the active regime given the true segmentation. Panel (c) considers a transfer setting where $w_2 = -w_1 + \varepsilon$ for small $\varepsilon$. Here, the optimal strategy is to leverage pre-change data via a Bayesian transfer prior rather than discard it (21). The known-in-advance transformer closely matches this baseline, indicating that it exploits inter-regime structure instead of ignoring pre-change observations. Panel (d) shows that the support-known transformer tracks the BMA baseline, which maintains a posterior over candidate change-point locations within the known support and averages predictions accordingly. This suggests that when partial information restricts but does not determine the change point, transformers learn to perform the corresponding hypothesis averaging in-context, consistent with Theorem 1.

## 4. Forecasting Infectious Diseases with Policy Changes

Recent advances in infectious disease forecasting have introduced foundation models that generalize across diseases and surveillance settings without retraining (24; 25), but

these models often struggle under regime shifts—such as those induced by government policy—due to their stationarity assumption. To study this, we use Thailand infectious disease data from 1980–2021, which includes multiple policy-driven shifts and provides a realistic testbed for non-stationarity, and evaluate whether encoding change-point information enables a pretrained model to adapt at inference time.

**Setup** We construct disease-policy episodes from Thai infectious disease data, each consisting of a window around a known policy change. Following the tabular formulation of TabPFN-TS (26), each episode is converted into a dataset with one row per time step, including the relative time index, z-score normalized value, a binary direction indicator (upward or downward shift), and a positional-encoding feature. We compare three variants: no PE, linear PE based on distance to the change point, and sinusoidal PE. Each episode has length 30. For the target sequence, the change point is fixed at $t = 12$; for context sequences, it is sampled uniformly from $10, \ldots, 20$, preventing reliance on a fixed transition time.

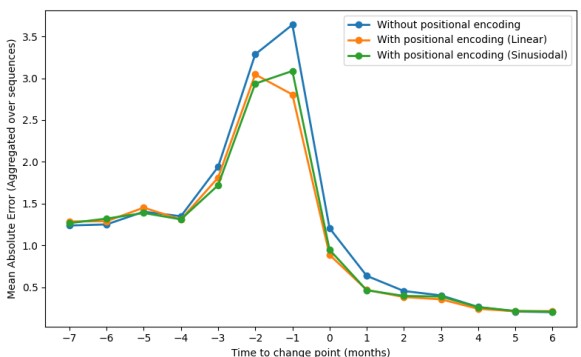

*Figure 2.* Mean absolute error for infectious disease forecasting versus forecast origin time relative to the policy change point. Each forecast is a recursive three-step-ahead prediction, so forecasts originating shortly before the change point span the regime transition. Linear and sinusoidal PEs reduce MAE by 25% at the most affected origins compared to no-PE.

**Evaluation and Results** We evaluate all time points $t$ in the target sequence. At each $t$, the model forecasts the next three observations, and we compute MAE over this horizon, averaged across episodes. We compare no-PE, linear-PE, and sin-PE settings. Figure 2 shows MAE as a function of forecast origin time relative to the change point. The largest gap between PE variants and the no-PE baseline occurs shortly before the change point, where the best PE variant reduces error by about 25%, since three-step-ahead forecasts at these points span the regime transition. Without positional information, the model extrapolates pre-change dynamics, while PE enables it to anticipate the shift. After the change point, all methods improve as post-change observations

accumulate, and the benefit of PE diminishes as the new regime becomes directly observable. Overall, encoding change-point information enables adaptation at inference time, with the largest gains near regime boundaries.

## 5. Forecasting Financial Series with Monetary Policy Events

Scheduled monetary policy announcements can introduce abrupt changes in financial time series, making forecasting more difficult around event dates. In particular, Federal Open Market Committee (FOMC) announcements influence both short-term interest rates and broader asset markets (27; 28). Because these events are known in advance, they provide a natural testbed for whether explicit event-time information improves forecasting.

**Setup** We study two daily financial series from FRED: the effective federal funds rate and the S&P 500 index (29; 30), paired with FOMC statement dates (31). For each business day $t$, the target is the next-day value. Each observation is represented in tabular form together with standard time-series covariates (lags 1–10, first difference, 5-day rolling mean and standard deviation). We compare three variants: a no-PE baseline, a linear PE model and a sinusoidal PE model encoding the signed business-day distance to the nearest FOMC event. Evaluation uses walk-forward forecasting with rolling training windows of up to three years and testing on successive 21-day blocks. A TabPFN regressor is trained and evaluated for each split.

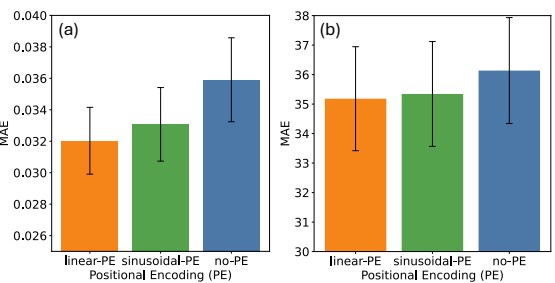

*Figure 3.* Mean absolute error for federal funds rate (a) and S&P 500 index (b) forecasting and under three feature settings: no PE, linear PE based on signed business-day distance to the nearest FOMC statement day, and sinusoidal PE. Error bars show the standard error.

**Evaluation and Results** We report overall MAE across all predictions, comparing three PE variants. Figure 3 shows a consistent ordering across both series: linear PE achieves the lowest MAE, followed by sinusoidal PE, with the no-PE baseline performing worst. This indicates that explicit event-time information helps the pretrained model adapt to event-related changes in financial dynamics.

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

# A. Full Proofs

This appendix provides complete proofs of the results stated in Section 2. We discuss preliminaries and give the full proof of Theorem 1 (Appendix A.1).

## A.1. Proof of Theorem 1

### A.1.1. NOTATION AND ASSUMPTIONS

We restate the model first. We observe $N$ input-label pairs generated by the piecewise-linear model

$$y_i = \begin{cases} \langle w_1, x_i \rangle + \varepsilon_i, & i \le n_1^*, \\ \langle w_2, x_i \rangle + \varepsilon_i, & i > n_1^*, \end{cases} \quad (1)$$

with inputs $x_i \in \mathbb{R}^d$ satisfying $\|x_i\| \le B_x$, labels satisfying $|y_i| \le B_y$, noise $\varepsilon_i \sim \mathcal{N}(0, \sigma^2)$, and prior $w_1, w_2 \overset{\text{iid}}{\sim} \mathcal{N}(0, \lambda^{-1} I_d)$. The change point $n_1^* \in \mathcal{K}_t \subseteq \{1, \ldots, t-1\}$ has uniform prior $\pi_k = 1/|\mathcal{K}_t|$ and candidate set size $|\mathcal{K}_t| \le R$.

**Sufficient statistics.** For each candidate change point $k \in \mathcal{K}_t$, define the post-change sufficient statistics

$$A_k(t) = \sum_{i=k+1}^{t-1} x_i x_i^\top, \quad b_k(t) = \sum_{i=k+1}^{t-1} x_i y_i, \quad c_k(t) = \sum_{i=k+1}^{t-1} y_i^2 \quad (4)$$

We also define the cumulative prefix sums

$$P_j^{xx} = \sum_{i=1}^{j} x_i x_i^\top, \quad P_j^{xy} = \sum_{i=1}^{j} x_i y_i, \quad P_j^{yy} = \sum_{i=1}^{j} y_i^2, \quad (5)$$

so that the segment statistics can be recovered by subtraction:

$$A_k(t) = P_{t-1}^{xx} - P_k^{xx}, b_k(t) = P_{t-1}^{xy} - P_k^{xy}, c_k(t) = P_{t-1}^{yy} - P_k^{yy}.$$

**Posterior under a fixed hypothesis.** For a fixed candidate $k$, the posterior over the post-change parameter $w$ is Gaussian:

$$w \mid Z_{<t}, k \sim \mathcal{N}(\mu_k, \Sigma_k),$$

where

$$\Sigma_k = (\lambda I + \sigma^{-2} A_k)^{-1}, \quad \mu_k = \sigma^{-2} \Sigma_k b_k. \quad (6)$$

The corresponding posterior predictive mean is

$$m_k(t) = x_t^\top \mu_k, \quad (7)$$

aka the ridge regression predictor computed on the post-change segment.

**Bayesian model averaging (BMA).** The BMA predictor maintains a posterior over candidate change points and forms a weighted prediction:

$$\hat{y}_t^{\text{BMA}} = \sum_{k \in \mathcal{K}_t} \alpha_k(t) \, m_k(t), \quad (8)$$

where the posterior weights satisfy

$$\alpha_k(t) \propto \pi_k \, p(Z_{<t} \mid k), \quad \sum_k \alpha_k(t) = 1,$$

and $p(Z_{<t} \mid k)$ denotes the marginal likelihood under hypothesis $k$.

**Bounded domain.** All quantities $(A_k, b_k, x_t)$ lie in a bounded set determined by $B_x, B_y$, and $N$, which ensures that the maps defined below are continuous on a compact domain.

### A.1.2. RESTATEMENT OF THEOREM 1

We restate the main approximation result in the notation introduced above.

**Theorem 2** (Transformer approximation of change-point BMA). *Consider the piecewise-linear model* (1) *with bounded inputs and the Gaussian prior and noise model specified above. Let $\hat{y}_t^{\text{BMA}}$ denote the Bayesian model-averaged predictor defined in* (8).

*For any $\epsilon > 0$, there exists a causal transformer $T_\epsilon$ such that, for all $t = 1, \ldots, N$ and all valid input sequences,*

$$\left| T_\epsilon(Z_{<t}, x_t) - \hat{y}_t^{\text{BMA}} \right| \le \epsilon. \quad (9)$$

*Moreover, the construction has the following properties:*

- *The number of layers is $O(1)$ for fixed $\epsilon$ and problem parameters.*

- *The number of attention heads scales as $O(|\mathcal{K}_t|)$.*

- *The hidden dimension scales as $O(|\mathcal{K}_t| \cdot d^2)$.*

The proof proceeds by constructing a transformer that (i) computes prefix sufficient statistics, (ii) isolates segment statistics for each candidate $k$, and (iii) approximates the BMA predictor (8) using MLP layers.

### A.1.3. TOKEN EMBEDDING

The initial representation of token $j$ is

$$h_j^{(0)} = \left[x_j;\ y_j;\ \mathrm{vec}(x_j x_j^\top);\ x_j y_j;\ y_j^2;\ \mathrm{PE}(j);\ \mathbf{0}\right] \in \mathbb{R}^D, \tag{10}$$

where $\mathrm{PE}(j)$ is a PE vector and $\mathbf{0}$ denotes unused coordinates reserved for intermediate computations in subsequent layers. The hidden dimension satisfies $D \geq d + 1 + d^2 + d + 1 + |\mathrm{PE}| + D_{\mathrm{work}}$, where $D_{\mathrm{work}}$ is the working space for intermediate computations.

We prove the theorem by explicit construction of a four-layer causal transformer that approximates the BMA predictor (2) to within $\epsilon$ on all valid input sequences. The construction proceeds layer by layer. We track approximation errors introduced at each stage and show that, by choosing MLP widths sufficiently large, the total error can be made smaller than any prescribed $\epsilon > 0$.

### LAYER 1: COMPUTING CUMULATIVE PREFIX SUMS

**Step 1a: Uniform causal averaging.** We configure a single attention head with $W_Q = 0$ and $W_K = 0$. Since all query-key inner products equal zero, the softmax attention weights reduce to the uniform distribution over the causal prefix:

$$\alpha_{ji} = \frac{1}{j}, \quad i = 1, \ldots, j. \tag{11}$$

Setting $W_V$ to be a projection onto the coordinates of $h_i^{(0)}$ containing $S_i^{\mathrm{raw}} := [\mathrm{vec}(x_i x_i^\top);\ x_i y_i;\ y_i^2]$ (which are included in the token embedding), the attention output at position $j$ is the running average

$$\bar{S}_j = \frac{1}{j} \sum_{i=1}^{j} S_i^{\mathrm{raw}}. \tag{12}$$

After the residual connection, the representation at position $j$ contains both $\bar{S}_j$ and $\mathrm{PE}(j)$.

**Step 1b: Recovering prefix sums via the MLP.** The cumulative prefix sum at position $j$ is $P_j = j \cdot \bar{S}_j$. Since $\bar{S}_j$ and the positional encoding $\mathrm{PE}(j)$ are both available in the residual stream, the MLP must approximate the map

$$(\bar{S}_j, \mathrm{PE}(j)) \mapsto j \cdot \bar{S}_j = P_j. \tag{13}$$

When $\mathrm{PE}(j) = j$ (linear PE), this is multiplication of two bounded scalars (applied entrywise), which is a continuous function on the compact domain $\{(\bar{s}, p) : |\bar{s}| \leq B_x^2 + B_x B_y + B_y^2,\ p \in \{1, \ldots, N\}\}$. When PE is sinusoidal, the map $j \mapsto \mathrm{PE}(j)$ is injective on $\{1, \ldots, N\}$. Thus there exists a function $g$ defined on its image such that $g(\mathrm{PE}(j)) = j$. The map $(\bar{S}_j, \mathrm{PE}(j)) \mapsto g(\mathrm{PE}(j)) \cdot \bar{S}_j$ is continuous on a bounded domain, and hence can be uniformly approximated by an MLP.

By the universal approximation theorem, for any $\epsilon_1 > 0$, there exists an MLP such that

$$\sup_{j=1,\ldots,N} \|\hat{P}_j - P_j\|_\infty \leq \epsilon_1.$$

The result is written to reserved coordinates in $h_j^{(1)}$ via the residual connection.

After Layer 1, the representation at each position $j$ contains $\hat{P}_j \approx P_j$ (the approximate prefix sums), the raw input $(x_j, y_j)$, the positional encoding $\mathrm{PE}(j)$, and the quadratic features $S_j^{\mathrm{raw}}$.

### LAYER 2: SEGMENT ISOLATION VIA RETRIEVAL

At time step $t$, the transformer must retrieve the prefix sums $\hat{P}_k$ at each candidate change-point index $k \in \mathcal{K}_t$ and form segment statistics by subtraction:

$$\begin{aligned}
\hat{A}_k(t) &= \hat{P}_{t-1}^{xx} - \hat{P}_k^{xx}, \\
\hat{b}_k(t) &= \hat{P}_{t-1}^{xy} - \hat{P}_k^{xy}, \\
\hat{c}_k(t) &= \hat{P}_{t-1}^{yy} - \hat{P}_k^{yy}.
\end{aligned} \tag{14}$$

**Step 2a: Multi-head retrieval of prefix sums.** For each candidate $k \in \mathcal{K}_t$, we assign a dedicated attention head $h$ with index $h = h(k)$. The number of heads required is $|\mathcal{K}_t| + 1$ (one per candidate plus one head to retrieve $\hat{P}_{t-1}$).

We configure head $h(k)$ so that, at the prediction position $t$, it attends strongly to position $k$. The query-key projections are designed over the PE coordinates:

$$W_Q^{(h)} h_t^{(1)} = \phi(\mathrm{PE}_{\mathrm{target}}(k)), \quad W_K^{(h)} h_j^{(1)} = \phi(\mathrm{PE}(j)), \tag{15}$$

where $\phi$ is chosen so that the vectors $\{\phi(\mathrm{PE}(j))\}_{j=1}^N$ are distinct and can be separated by inner products, i.e., for each $k$,

$$\langle \phi(\mathrm{PE}(k)), \phi(\mathrm{PE}(k)) \rangle > \langle \phi(\mathrm{PE}(k)), \phi(\mathrm{PE}(j)) \rangle \quad \text{for all } j \neq k.$$

The mechanism depends on how the candidate set $\mathcal{K}_t$ is communicated:

- **Support known** ($\mathcal{K}_t = \{L, \ldots, U\}$): The PE encodes the boundaries $L, U$. We configure $R = U - L + 1$ heads, with head $h$ having a query projection that targets position $L + h - 1$. Because PEs are distinct across positions, the query-key inner product can be made to peak at the target position. More concretely, for linear PE where $\mathrm{PE}(j) = j$, we set $W_Q^{(h)} = [0; \ldots; 1; \ldots; 0]$ selecting the coordinate encoding target position $k_h$, and $W_K^{(h)}$ selecting the PE coordinate of $h_j^{(1)}$. Scaling these projections by a large constant $C$ makes the attention weights $\alpha_{t,j}^{(h)}$ approximate $\mathbf{1}[j = k_h]$ as $C \to \infty$,

so the head retrieves $\hat{P}_{k_h}$ up to softmax leakage that is exponentially small in $C$.

- **Known exactly** ($\mathcal{K}_t = \{n_1^*\}$): A single head suffices, with the query keyed to the PE encoding of $n_1^*$.

- **No information** ($\mathcal{K}_t = \{1, \dots, t-1\}$): In principle, $O(t)$ heads are required. In practice, the training distribution restricts the effective support (e.g., $n_1^* \in \{N/2 - R, \dots, N/2 + R\}$), so $O(R)$ heads suffice for the model to implicitly learn the support.

The value projection $W_V^{(h)}$ selects the coordinates of $h_j^{(1)}$ containing $\hat{P}_j$. After the multi-head attention, the residual stream at position $t$ contains approximate copies of $\hat{P}_k$ for each $k \in \mathcal{K}_t$, as well as $\hat{P}_{t-1}$ (from either a dedicated retrieval head or the self-retrieval at position $t-1$ via the residual).

**Retrieval accuracy.** Let $\alpha_{t,j}^{(h)}$ denote the attention weights of head $h$ at position $t$. With query-key scaling $C$, the weight on the target position $k_h$ satisfies

$$\alpha_{t,k_h}^{(h)} = \frac{\exp(C)}{\exp(C) + \sum_{j \neq k_h} \exp(s_j)} \geq 1 - (t-1)e^{-C + S_{\max}}, \tag{16}$$

where $s_j$ are the non-target scores and $S_{\max} = \max_{j \neq k_h} s_j$. Choosing $C$ large enough (dependent on $N$ and $\epsilon$ but not on the data), the retrieved value satisfies $\|\text{Attn}^{(h)}(t) - \hat{P}_{k_h}\| \leq \epsilon_{\text{ret}}$ for any prescribed $\epsilon_{\text{ret}} > 0$.

**Step 2b: Forming segment statistics.** The Layer 2 MLP receives $\hat{P}_{t-1}$ and the retrieved values $\hat{P}_k$ for $k \in \mathcal{K}_t$, and computes segment statistics by subtraction (14). Subtraction is a linear operation implemented exactly by the MLP (or even by a linear layer). The approximation error in the segment statistics inherits from the prefix sum errors and retrieval error:

$$\|\hat{A}_k(t) - A_k(t)\|_F \leq 2\epsilon_1 + \epsilon_{\text{ret}}, \tag{17}$$

and similarly for $\hat{b}_k$ and $\hat{c}_k$. Define $\epsilon_2 := 2\epsilon_1 + \epsilon_{\text{ret}}$.

After Layer 2, the residual stream at the prediction position $t$ contains $(\hat{A}_k, \hat{b}_k, \hat{c}_k)$ for every $k \in \mathcal{K}_t$, the query input $x_t$, and the PE. All subsequent computation occurs only at position $t$, involving no further attention across positions. This ensures causality without backward information flow.

LAYERS 3–4: POSTERIOR COMPUTATION AND PREDICTION

**Step 3: Predictive means.** Define the domain

$$\mathcal{S} := \left\{ (A, b, x) : A \succeq 0, \ \|A\|_F \leq NB_x^2, \ \|b\| \leq NB_x B_y, \ \|x\| \leq B_x \right\}$$

For each candidate $k \in \mathcal{K}_t$, the posterior predictive mean under the Gaussian model is Ridge regression, i.e.,

$$m_k(t) = x_t^\top \mu_k, \quad \text{where} \quad \mu_k = \sigma^{-2}(\lambda I + \sigma^{-2} A_k)^{-1} b_k. \tag{18}$$

The map $\psi_m : (A_k, b_k, x_t) \mapsto m_k(t)$ is continuous on the compact domain $\mathcal{S}$. In fact, $\psi_m$ is Lipschitz on $\mathcal{S}$. To see this, note that $(\lambda I + \sigma^{-2} A)^{-1}$ is Lipschitz in $A$ on the set $\{A \succeq 0 : \|A\|_F \leq NB_x^2\}$ because

$$\|M_1^{-1} - M_2^{-1}\| \leq \|M_1^{-1}\| \cdot \|M_2^{-1}\| \cdot \|M_1 - M_2\| \leq \lambda^{-2}\|M_1 - M_2\| \tag{19}$$

for $M_i = \lambda I + \sigma^{-2} A_i \succeq \lambda I$. The remaining operations (matrix-vector products, inner products) are bilinear and hence Lipschitz on bounded domains. Composing, there exists a constant $L_m$ depending on $B_x, B_y, \lambda, \sigma, N, d$ such that

$$|m_k(t; \hat{A}_k, \hat{b}_k) - m_k(t; A_k, b_k)| \leq L_m \cdot \epsilon_2. \tag{20}$$

By the universal approximation theorem, an MLP with two hidden layers can approximate $\psi_m$ on $\mathcal{S}$ to within any $\epsilon_3 > 0$, using width that depends on $\epsilon_3$, $d$, and the constants $B_x, B_y, \lambda, \sigma$. The MLP computes all $|\mathcal{K}_t|$ maps in parallel by using $|\mathcal{K}_t|$ independent subnetworks within its width. The total error in the predictive mean for candidate $k$ is at most $L_m \epsilon_2 + \epsilon_3$.

**Step 4a: Log-marginal likelihoods.** The log-marginal likelihood under hypothesis $k$ is:

$$\ell_k := \log p(Z_{<t} \mid k) = -\frac{n_k}{2} \log(2\pi\sigma^2) \\ -\frac{1}{2} \log \det(\lambda^{-1} A_k + I) \tag{21} \\ -\frac{1}{2\sigma^2}\left(c_k - \sigma^{-2} b_k^\top \Sigma_k b_k\right),$$

where $\Sigma_k = (\lambda I + \sigma^{-2} A_k)^{-1}$ and $n_k = t - 1 - k$ is the number of post-change samples under hypothesis $k$. The first term depends only on $n_k$ (which is determined by $k$ and $t$, both available through the PE and the position in the sequence). The second and third terms are continuous functions of $(A_k, b_k, c_k)$ on $\mathcal{S}$ by Lemma **??**(ii).

The map $\psi_\ell : (A_k, b_k, c_k, n_k) \mapsto \ell_k$ is again Lipschitz on $\mathcal{S}$, with Lipschitz constant $L_\ell$ depending on the problem parameters. In particular:

- The log-determinant satisfies $|\log \det(M_1) - \log \det(M_2)| \leq d \cdot \|M_1^{-1}\| \cdot \|M_1 - M_2\|$ for nearby positive definite matrices $M_1, M_2 \succeq I$, so the log-determinant term is Lipschitz in $A_k$ with constant depending on $d$.

- The quadratic form $b_k^\top \Sigma_k b_k$ is Lipschitz in $(A_k, b_k)$ by the same argument as (19) combined with bilinearity.

- The term $c_k$ enters linearly.

Therefore $|\ell_k(\hat{A}_k, \hat{b}_k, \hat{c}_k) - \ell_k(A_k, b_k, c_k)| \le L_\ell \cdot \epsilon_2$.

An MLP approximates $\psi_\ell$ on $\mathcal{S}$ to within $\epsilon_4 > 0$ for each of the $|\mathcal{K}_t|$ candidates. Total error in each log-marginal likelihood: $L_\ell \epsilon_2 + \epsilon_4$.

**Step 4b: Posterior weights and weighted prediction.**
The posterior weights are $\alpha_k(t) = \mathrm{softmax}_k(\ell_1, \ldots, \ell_{|\mathcal{K}_t|})$. The softmax function is Lipschitz on bounded domains: for $\ell, \ell' \in [-M, M]^R$,

$$\|\mathrm{softmax}(\ell) - \mathrm{softmax}(\ell')\|_1 \le 2\|\ell - \ell'\|_\infty. \quad (22)$$

(This follows from the fact that each partial derivative of softmax is bounded by 1.) Let $\hat{\ell}_k$ denote the MLP's approximation to $\ell_k$. The error in $\hat{\ell}_k$ is at most $L_\ell \epsilon_2 + \epsilon_4$, so the posterior weight error is bounded by

$$\|\hat{\alpha} - \alpha\|_1 \le 2(L_\ell \epsilon_2 + \epsilon_4). \quad (23)$$

Finally, the BMA prediction is the weighted sum $\hat{y}_t^{\mathrm{BMA}} = \sum_k \alpha_k m_k$. The MLP computes $\hat{y}_t = \sum_k \hat{\alpha}_k \hat{m}_k$. The total error decomposes as

$$\left|\hat{y}_t - \hat{y}_t^{\mathrm{BMA}}\right| = \left|\sum_k \hat{\alpha}_k \hat{m}_k - \sum_k \alpha_k m_k\right|$$

$$\le \left|\sum_k (\hat{\alpha}_k - \alpha_k)\hat{m}_k\right| + \left|\sum_k \alpha_k(\hat{m}_k - m_k)\right|$$

$$\le \|\hat{\alpha} - \alpha\|_1 \cdot \max_k |\hat{m}_k| + \max_k |\hat{m}_k - m_k|, \quad (24)$$

where we used $\sum_k |\alpha_k| = 1$ and $\sum_k |\hat{\alpha}_k - \alpha_k| = \|\hat{\alpha} - \alpha\|_1$. The predictive means are bounded: $|m_k| \le B_x \cdot \|\mu_k\| \le B_x \lambda^{-1} \sigma^{-2} N B_x B_y =: M_m$, and the approximate means satisfy $|\hat{m}_k| \le M_m + L_m \epsilon_2 + \epsilon_3$.

Combining all error terms:

$$\left|\hat{y}_t - \hat{y}_t^{\mathrm{BMA}}\right| \le 2(L_\ell \epsilon_2 + \epsilon_4)(M_m + L_m \epsilon_2 + \epsilon_3) + (L_m \epsilon_2 + \epsilon_3). \quad (25)$$

**Choosing tolerances.** Given $\epsilon > 0$, we choose:

1. $\epsilon_1$ (prefix sum MLP error) small enough,

2. $\epsilon_{\mathrm{ret}}$ (retrieval accuracy, controlled by the query-key scaling $C$) small enough,

3. $\epsilon_3$ (predictive mean MLP error) small enough,

4. $\epsilon_4$ (log-marginal likelihood MLP error) small enough,

such that $\epsilon_2 = 2\epsilon_1 + \epsilon_{\mathrm{ret}}$ and the total error (25) is at most $\epsilon$. Since (25) is a polynomial in $(\epsilon_1, \epsilon_{\mathrm{ret}}, \epsilon_3, \epsilon_4)$ with no constant term, this can always be achieved by choosing each tolerance sufficiently small (with the required smallness depending on the problem parameters $B_x, B_y, \lambda, \sigma, N, d$).

**Resource accounting.** The construction uses:

- **Layers:** 4 (one for prefix sums, one for retrieval and subtraction, two for posterior computation). The number of layers is $O(1)$ for fixed $\epsilon$ and problem parameters.

- **Attention heads:** Layer 1 uses 1 head (uniform averaging). Layer 2 uses $|\mathcal{K}_t| + 1 \le R + 1$ heads. Layers 3–4 require no attention (MLP-only computation at the prediction token). Total: $O(R)$ heads.

- **Hidden dimension:** Must accommodate $d^2 + d + 1$ coordinates for each of the $|\mathcal{K}_t|$ candidates' sufficient statistics, plus $d$ coordinates for $x_t$, plus working space for the MLP computations. Total: $O(R \cdot d^2 + D_{\mathrm{MLP}})$, where $D_{\mathrm{MLP}}$ depends on $\epsilon$ through the MLP width needed for universal approximation.

This completes the proof. $\square$