# OpenReview forum: "In-Context Learning Under Regime Change"
_ICML.cc/2026/Workshop/FMSD — FMSD @ ICML 2026 Poster_

### Official Review · Reviewer_oDw9 · 2026-05-20
**Clean theory and convincing validation on synthetic and real-world data, with minor concerns.**

**Rating:** 8
**Confidence:** 4

**Review:**

**Summary**: The paper formalizes in-context change-point detection as a framework for how transformers adapt to non-stationary sequences. Its theoretical result is that a transformer can approximate the Bayes-optimal predictor (Bayesian model averaging) for a piecewise-linear regression with a single unknown change point, with positional encoding carrying the side information about the change-point location. Empirically, transformers trained on synthetic piecewise-linear data match these optimal Bayesian baselines across information levels, with performance improving as more information is provided. The authors also add a positional-encoding feature to a frozen TabPFN style model for both infectious-disease and financial (FOMC) forecasting, lowering MAE in each, with the disease improvements concentrated near the change point.

**Strengths**:
1. Clean problem framing. Formalizing transformer adaptation to non-stationarity as in-context change-point detection, with a four-level information hierarchy (no information, support known, known in advance, known afterward), gives a clear way to study how side information changes the difficulty of adaptation, and it fits the workshop theme on foundation models for structured data.

2. The layer-by-layer approximation of the Bayesian model-averaging predictor is explicit, tracks approximation error at each stage, and combines Lipschitz arguments with universal approximation. The capability and complexity tradeoff, where the information level sets width and head count while depth stays constant, is a clean characterization.

3. Synthetic results matching theory/intuition. Across information levels the trained transformers track their matched Bayesian baselines, and the ordering of the variants follows intuition, with the known-in-advance variant achieving the lowest error and recovering fastest at the transition.

4. Empirical success in transferring intuition to real-world demonstration. Even without retraining, supplying change-point information to a frozen TabPFN style forecaster lowers error on both infectious-disease and financial series, and on the disease data the largest gains concentrate near regime boundaries. That the improvement is strongest exactly where adaptation matters is evidence the signal aids adaptation rather than acting as a generic covariate.

**Areas for Improvement:**

1. The theory is cleanly scoped to a single regime change in a piecewise-linear model with Gaussian noise, and the paper is upfront about these assumptions. Real series are usually nonlinear and pass through several regime changes, so how the approach generalizes beyond this synthetic setting is the natural and genuinely interesting open question. The trained-specialist direction in the next point is one promising route toward it.

2. The theory and the synthetic experiments train a transformer with the change point carried in the embedding, but the real-world study drops this and instead adds a change-point feature to a frozen TabPFN. Encoding a known change point as a feature and watching a forecasting model improve is routine in applied data science, where the same move helps gradient boosting and other tabular models, so presenting it as a new result overstates its novelty. The more compelling and genuinely novel direction, and the one that would connect back to the theory, is to fine-tune or train a TabPFN with the change-point embedding so the model learns to exploit change-point structure, then compare that specialized model against the plain-feature and in-context baselines.

3. The work is framed as largely new territory, yet the intersection of change-point detection, both linear and nonlinear, with transformers and large language models is already drawing attention, including in finance. The paper would be stronger if it positioned itself at this intersection and contrasted foundation-model approaches with classical change-point methods. Specific references are left to the authors. The classical online and offline change-point literature is likely already cited, so the more relevant gap is recent deep-learning, LLM-based, and agent-based change-point detection, together with foundation models for non-stationary time series.

**Detailed Comments:**

- Clarify that four layers is sufficient rather than necessary, and that the substantive claim is a construction of constant depth independent of the approximation accuracy.

- The boundedness assumption on labels and inputs cannot hold almost surely under Gaussian noise, since there is a non-zero probability that $y_i$ escapes any fixed bound, which conflicts with the distributional assumption on the $\varepsilon_i$. Consider truncated noise (a truncated normal is common in Bayesian analysis) or arguing boundedness in expectation and upgrading the proof. Minor and fixable for a workshop, but worth making rigorous if the authors target a main conference.

- For the real-world experiments, describe the positional-encoding features and covariates precisely, ideally in a short appendix or released code, and clarify whether the financial within-block forecasts use realized or predicted lags and whether the reported error is one-step-ahead pooled across rolling blocks. These details are needed to make the results both reproducible and unambiguous.

- Add a short future-work section. Natural directions include training or fine-tuning a change-point-aware specialist, handling multiple change points, and modeling nonlinear or non-Gaussian regimes. Even a paragraph of whatever the authors find most promising would help readers gauge the framework's reach.

**Justification of Score:**

I am giving this paper an 8 out of 10. It poses a clean and timely question and answers it convincingly on the synthetic side, with a careful theoretical construction and experiments that match intuition across every information level. This is also where the paper fits the workshop best. The workshop centers on in-context learning and foundation models for structured and time-series data, and asks whether purpose-built models outperform general-purpose ones, so a transformer trained from scratch with a change-point encoding, backed by theory that explains why it recovers the Bayes-optimal predictor, is exactly the kind of specialized structured-data model the call is about.

The reason it is not a 9 or 10 is that this mechanism does not carry over to the real-world experiments. There the change point is added as a feature to a frozen TabPFN rather than trained into the model, so the most compelling version of the contribution, a change-point-aware foundation model with new weights, is never shown on real data. It would be genuinely interesting to see a TabPFN trained from scratch, or fine-tuned, to detect and adapt to change points, and that is the result that would lift the paper.

My confidence is 4 out of 5, as I have read the paper in depth, including a brief look at the theorem in the appendix (though I did not check every single detail in the proof of the theorem itself).

---

### Official Review · Reviewer_PyK6 · 2026-05-20
**This reviewer believes there are insufficient results to justify the inclusion of this work.**

**Rating:** 4
**Confidence:** 4

**Review:**

## Summary:
&emsp;Authors investigate positional encoding and the incorporation of changepoint timing information into a transformer model. They provide a theoretical argument that causal transformers approximate Bayesian model-averaged predictors and quantify model performance on synthetic problems
Strengths:
&emsp;I think the theoretical portion of this work is its strongest.
## Areas Of Improvement:
&emsp;Presentation of results needs to include a measure of uncertainty/variability, given the synthetic evaluation is done over 5000 test trajectories inclusion of the standard deviation in some form should be done. Additionally the use of a constant fixed change timepoint makes it hard to truly evaluate the impact on model performance as testing is so well within the training distribution. I would suggest an evaluation which tests performance on an increasingly distant change timepoint to more robustly evaluate performance.

&emsp;Additional experimental results comparing the MSE/MAE with other similar models would better contextualize this work's contribution. As it stands I have no sense of what the baseline for the field is
Detailed Comments:

&emsp;Rhetoric used when describing the results is far too confident. Authors claim that they “solve” the in-context change point detection/adaptation problem. The experimentation and real world evaluation is nowhere near robust enough to justify such a claim.
## Justification for Score:
&emsp;The work presented is too sparse and not well enough justified by either its results or potential for impact to recommend its inclusion.